# Aetiopathogenesis of Rotator Cuff Tear in Patients Younger than 50 Years: Medical Conditions Play a Relevant Role

**DOI:** 10.3390/medicina59050998

**Published:** 2023-05-22

**Authors:** Stefano Gumina, Lorenzo Mezzaqui, Rossana Aimino, Marco Rionero, Alessandra Spagnoli, Vincenzo Campagna, Vittorio Candela

**Affiliations:** 1Department of Anatomical, Histological, Forensic Medicine and Orthopaedics Sciences, Sapienza University of Rome, 00100 Rome, Italy; 2Icot Latina, 00100 Latina, Italy; 3Department of Public Health and Infectious Diseases, Sapienza University of Rome, 00100 Rome, Italy; 4Orthopaedics and Traumatology Unit, Celio Military Hospital, 00100 Rome, Italy

**Keywords:** rotator cuff tear etiology, rotator cuff tear in young patients, comorbidity in rotator cuff tear, obesity and rotator cuff tear, smoking habit and rotator cuff tear

## Abstract

*Background and Objectives:* Studies on rotator cuff tears (RCT) in patients younger than 50 years have focused on the post-operative outcomes. Little is known about cuff tear etiopathogenesis, although it is a common belief that most tears are due to trauma. We have retrospectively verified the prevalence of medical conditions, whose role in tendon degeneration development have been widely demonstrated, in a group of patients younger than 50 years with postero-superior RCT. *Materials and Methods*: 64 patients [44M-20F; mean age (SD): 46.90 (2.80)] were enrolled. Personal data, BMI, smoking habit, diseases (diabetes, arterial hypertension, hypercholesterolaemia, thyroid diseases, and chronic obstructive pulmonary disease) were registered. The possible triggering cause and the affected side and tear dimensions were recorded, and statistical analysis was then performed. *Results*: 75% of patients had one or more diseases and/or a smoking habit for more than 10 years. In the remaining 25%, only four patients referred had had a traumatic event, while in the other eight patients, both medical condition and trauma were registered. The presence of two or more diseases did not affect RCT size. *Conclusions*: In our series, three quarters of patients with RCT had a smoking habit or medical conditions predisposing them to a tendon tear; therefore, the role of trauma in RCT onset in patients younger than 50 years is markedly resized. It is plausible that in the remaining 25%, RCT may be due to trauma or to genetic or acquired degeneration. Level of Evidence: IV

## 1. Introduction

The scientific interest in rotator cuff tear (RCT) in patients younger than 50 years is mainly focused on clinical outcomes after repair [1,2,3,4,5,6,7,8,9,10,11,12]; on the other hand, etiopathogensis has aroused little attention. It is a common belief that, in these patients, RCT predominantly occurs following trauma. In 2014, MacKechnie et al. [2], in a systematic review, selected seven studies [5,6,7,9,10,11,12] that examined the etiopathogenesis of the full-thickness RCT and the results obtained after open or arthroscopic repair in patients aged younger than 55 years old. Eighty-one percent (from 60% to 100%) of the RCT were attributed to a trauma. Twenty years earlier, in another systematic review, Lazarides et al. [1] concluded that in young patients (range 16–40 years), RCTs are mainly due to traumatic tears and, in a smaller proportion, to elite throwing.

The traumatic hypothesis contrasts with two scientific evidences: (a) the young’s cuff tendon tissue is more resistant than that of the elderly [13] and therefore should not undergo degeneration and tear following a simple fall, and (b) recent studies mainly attribute the responsibility for tendon tears to systemic diseases or habits that might alter the peripheral microcirculation and cause tissue hypoxia, with consequent cellular apoptosis [14,15,16,17,18,19,20,21,22].

We have retrospectively verified the prevalence of medical conditions, which alter peripheral microcirculation, in a population with full-thickness posterosuperior RCT in order to resize the potential role of trauma in the genesis of RCT in young patients.

## 2. Materials and Methods

We have collected medical records of patients younger than 50 years who underwent arthroscopic repair for a postero-superior RCT between September 2017 and September 2019.

Anagraphycal data, BMI, smoking habit, medical conditions (diabetes, arterial hypertension, hypercholesterolaemia, thyroid diseases, and chronic obstructive pulmonary disease) and any previous surgical procedures were registered. Finally, the affected side and tear dimension were noted. In three patients, medical records did not provide detailed information, so they were contacted by phone. Patients with past or current cancer, rheumatoid arthritis, osteoarthritis, pigmented villonodular synovitis, and psychiatric conditions were excluded. We also excluded patients with upper third subscapularis tendon tear (n.2) and partial thickness posterosuperior tears, and no complete subscapularis tears were found in our series.

The Southern California Orthopedic Institute classification of complete rotator cuff tears [23] was used to classify tendon tears intraoperatively as follows: (1) complete tears, such as a puncture wound (type I); (2) tears usually <2 cm that still encompassed only 1 of the rotator cuff tendons, with no retraction of the torn ends (type II); (3) complete tears involving an entire tendon, with minimal retraction of the torn edge, usually 3–4 cm (type III); and (4) tears involving 2 or more rotator cuff tendons, often with associated retraction and scarring of the remaining tendon ends, and often an L-shaped tear that is frequently irreparable (type IV). To limit the number of groups and to make the sample more representative, we considered the lesions belonging to type I to be small, those of types II and III to be large, and those of type IV to be massive.

All individuals were well informed on the purpose of the research; that data would have been stored and anonymously disclosed; that the research would not have been economically financed; and that the authors would not have received compensation or other benefits.

Our sample was compared to data taken from epidemiologic surveys [24,25,26,27] lead by the National Institute of Health belonging to a same age-matched population.

According to the law of our country, this study does not need Ethical Committee approval.

## 3. Statistical Analysis

Patient characteristics were described using median with 25th and 75th percentiles for continuous variables and percentages for dichotomous variables. Chi-square and Kruskall–Wallis tests were used to study the association between tear size and patient characteristics. Statistical analysis was performed using R version 3.6.1.

## 4. Results

The final study group was composed of 64 cases [44M-20F; mean age (SD): 46.90 (2.80)]. Baseline characteristics, relative to tear size, side, and mechanism of lesion, are shown in Table 1.

Intraoperatively, none of our patients had evident ecchymosis on the torn tendon edges. In Table 2, a consecutive series of 64 patients with different sized RCTs, information regarding body mass index, smoking habit, and medical conditions are listed. RCT: in the table stands for rotator cuff tear.

No significant correlation was found (*p* < 0.05) between tear size and age, mechanism of lesion, smoking habit, and BMI (Table 3). The chi square test was applied to evaluate the correlation between tear size and age, mechanism of lesion, smoking habit, and BMI (Table 3). No significant correlation was found (*p* < 0.05).

According to the smoking habit, 56.2% (36 patients: 30M-6F) were smokers. Twenty-four patients (37.5%; 22M-2F) reported a daily smoking of more than 10 cigarettes for a minimum of 10 years; among them, 9.4% smoked more than 20 cigarettes every day.

In our cohort, 62.5% of patients had a BMI greater than 25. In particular 16 patients (25%; 12M-4F) were obese (10 patients with grade I obesity, 4 with grade II and 2 with severe obesity).

Two male patients suffered from Type I diabetes. They developed a large RCT with no history of trauma.

Ten patients (16.1%; 6M-4F) with RCT were found to suffer from arterial hypertension for more than 2 years.

Hypercholesterolemia was found in 9.4% (4M-2F) of patients; oral therapy was assumed for more than 5 years.

Relative to thyroid disease, 9.4% (6F) of patients suffered from hypothyroidism treated with levothyroxine. Finally, a chronic obstructive pulmonary disease was present in six male patients (9.4%). Considering the whole group, 75% of patients (48/64) had one or more medical conditions (obesity, diabetes, arterial hypertension, hypercholesterolemia, thyroid disease, and chronic obstructive pulmonary disease) and/or they smoked for more than 10 years. In the remaining 25% (16/64), only eight patients sustained a RCT due to a traumatic event, while in the other eight patients, no diseases and trauma were registered.

Figure 1 summarized the prevalence of BMI, smoking habit, diabetes, arterial hypertension, hypercholesterolemia, thyroid disease, and copd between our cohort and the same age-matched control population in our country.

## 5. Discussion

Our study revealed that postero-superior RCTs in young patients are not caused exclusively by trauma but medical conditions play a significant role.

Prior to our findings, it was a common belief that RCT occurred in young patients following a traumatic event. In 2004, Sperling et al. [12] examined the clinical outcomes and the failures rates of open cuff repair in patients aged 50 and younger. Of 32 patients, 7 (22%) died after less than thirteen years of follow-up. Similarly, in the Burns and Snyder series [4], composed of 52 patients younger than 50 years of age, 2 patients (4%) died during an average follow-up of 5.8 years, and, furthermore, 13 were lost at the follow-up. In both studies, the death rate is too high. In a statistical survey conducted in the United States in 2017 [25], it emerged that in the male population between 50 and 54 years and between 55 and 59 years of age, the percentage of deaths is 0.60% and 0.91%, respectively, while in the female population, it is 0.37% and 0.56%. Neither of the two studies refer to the general health conditions of deceased patients, to their BMI, or to their causes of death. However, it can be plausible that the dead patients had important medical conditions. Furthermore, the two previous papers [4,12] show that of the 25 and 37 patients clinically evaluated at the follow-up, 5 (20%) and 4 (11%), respectively, had a bilateral RCT. Again, it is highly improbable that cuff tears occurred in both shoulders following trauma.

In the Dwyer et al. [5], Sperling et al. [12] and Hawkins et al. [6] series relating to young patients with RCTs, the prevalence of large and massive tears was 16.6%, 37.9% and 42.1%, respectively. In our series, the prevalence was 53.1%. No scientific evidence is present between the type of trauma (simple fall, sport injury, road-traffic accident, fall from height, and direct frontal blow) and RCT size; however, it is questionable that a massive tear may occur as a consequence of a simple fall. In this regard, Lin et al. [9] reported that of the 32 young patients who sustained a rotator cuff tear following trauma, 8 (25%) reported a simple fall. In our studied group, 30 patients (46.8%) correlated rotator cuff tear onset to a trauma, and 20 of them (31.2%) claimed that shoulder pain arose following a simple fall. A total of 6 (75%) of our 8 patients with a massive rotator cuff tear denied any trauma.

In 2010, Baumgarten et al. [28] observed a strong association between smoking habit and rotator cuff disease. Carbone et al. [17] observed that there is a correlation between the number of smoked cigarettes, rotator cuff tear, and size. The prevalence of smokers in our cohort is more than double with respect to the general population between 45 years and 64 years (24.6%) registered by epidemiologic surveys in our country [24].

Individuals with type II diabetes mellitus are four times more likely to experience tendinopathy, and up to five times more likely to experience a tendon tear than non-diabetics [29]. Thomas et al. [30] reported a 27.5% incidence of shoulder disorders in Type 2 diabetes mellitus patients (vs. 5% in non-diabetics) while Lin et al. [30] identified a greater than twofold increase in rotator cuff disorders in diabetic patients. In our studied cohort, no patients suffered from Type 2 diabetes, and only two patients suffered from type I diabetes.

In a frequency-matched case-control study, Wendelboe et al. [22] observed that increasing BMI is a risk factor for rotator cuff pathology. Results of a case-control design study [15] provided evidence that obesity, measured through BMI and percentage of body fat, is a significant risk factor for both RCT occurrence and severity. In our cohort, 62.5% of patients had a BMI greater than 25%, and 25% were obese. This data is enormously higher with respect to the percentages reported in our official annual country reports relative to obesity in subjects aged between 45 and 54 years: BMI > 30 is present in 13.8% of males and 8.8% of females [24].

In a case-control study [14], hypertension was found to be associated with a twofold higher risk of cuff tear occurrence and severity. In our series, 10 patients (16.1%) suffered from arterial hypertension for more than 2 years. This prevalence is similar to that of our national register of arterial hypertension [27] for the same age-based population group (20.8%).

In 2010, Abboud and Kim [31] prospectively collected serum cholesterol and lipid profiles on patients with RCT and on controls. Patients with RCTs were found to have higher levels of cholesterol. Hypercholesterolemia may alter the tendon’s extracellular matrix and impair macro- and microcirculation [32], increasing degeneration rates or preventing healing [33]. In our series, hypercholesterolemia was found in 9.4% of patients. This value is higher than that reported in our national register on dyslipidemias (6% of peers subjects) [24].

The role of thyroid hormones in the homeostasis of shoulder tendons arouses interest; a recent study, performed on human tenocytes isolated from cuff tendons, has demonstrated the presence of thyroid hormone receptors in healthy and pathologic tendons, thus suggesting a possible role in the proliferation and apoptosis of human tenocytes [34]. Vicenti et al. [35] concluded that thyroid diseases should be considered as risk factors for shoulder pain. In our studied group, 6 patients (9.4%) suffered for hypothyroidism assuming levothyroxine. This prevalence is three times higher than that reported in our national register on thyroid diseases (3.14% of peers subjects) [24].

It is well known that patients with copd have a reduced oxygen saturation; therefore, it cannot be excluded that peripheral tissue oxygenation, including that of the rotator cuff, may be compromised. Unfortunately, this suggestive hypothesis has aroused little interest. In our series, copd was present in 6 male patients (9.4%). Once again, this percentage is higher than that reported by the General Medicine Care Society of our Country [24]. In fact, according to a recent survey, copd is present in only 2% of subjects aged between 45 and 54 years.

A total of 75% of our patients had one or more medical conditions (including those related to the smoking habit) whose role in determining RC degeneration and tear has been widely demonstrated. The presence of two or more diseases did not affect RCT size; in fact, among the 22 patients with many medical conditions, 10 had a small tear (average age 46.4 years), 10 a large tear (average age 47.6 years), and 2 had a massive tear (49 years). Among the 16 patients with no medical conditions, only 8 (12.5%) referred a trauma; for the other 8 (12.5%), however, cuff tear etiology remains uncertain and, therefore, might be attributable to tendon degeneration due to genetic or acquired causes [36,37].

Still, the present study has some limitations that have to be addressed. Partial thickness tears were not considered, to name one, and the control group was not enrolled but a high-numerosity database with updated information was used.

## 6. Conclusions

The role of trauma in the genesis of RCT in patients younger than 50 years is markedly resized; in three quarters of cases, a smoking habit or medical conditions, predisposing to tear, were present. In the remaining 25%, RCT can be attributed to trauma or genetic or acquired degeneration.

## Figures and Tables

**Figure 1 medicina-59-00998-f001:**
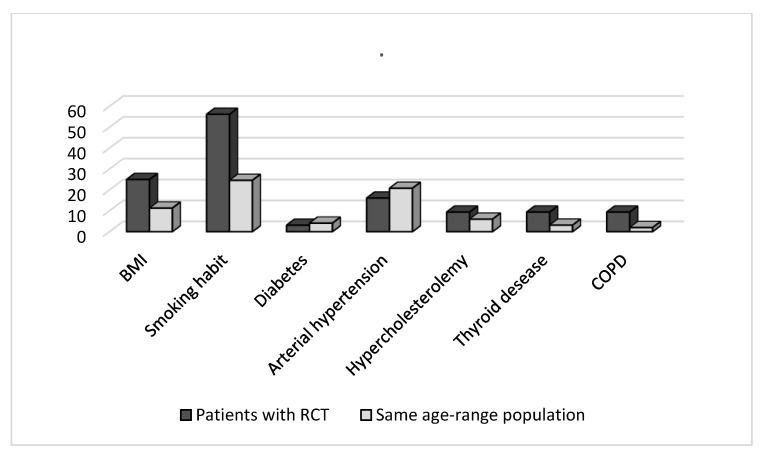
Prevalence of smoking habit, BMI, diabetes, arterial hypertension, hypercholesterolemia, thyroid disease, and chronic obstructive pulmonary disease (COPD) between our cohort and a same age-matched control population.

**Table 1 medicina-59-00998-t001:** Baseline characteristics of the studied group, relative to the rotator cuff tear side, mechanism of lesion, and size. RCT: rotator cuff tear.

	Patients (n: 64)
RCT Side	Dominant shoulder: 28 cases (43.7%)
	Non-dominant shoulder: 36 cases (56.3%)
Mechanism of lesion	Traumatic: 30 (46.8%—22M-8F)
	Simple fall while walking: 20 (66.6%) Fall during sport activity: 8 (26.6%) Car accident: 2 (6.8%)
	Atraumatic: 34 (53.2%—22M-12F)
RCT Size	Small: 30 (46.9%) Large: 26 (40.6%) Massive: 8 (12.5%)

**Table 2 medicina-59-00998-t002:** Lists of a consecutive series of 64 patients with different sized RCTs, with information regarding body mass index, smoking habit, and medical conditions.

			Rotator Cuff Tear					Medical Conditions				
**PATIENT**	**GENDER**	**AGE AT SURGERY**	**DOMINANT SIDE**	**MECHANISM**	**TEAR SIZE**	**BMI**	**SMOKING HABIT**	**DIABETES**	**ARTERIAL HYPERTENSION**	**HYPERCHOLE STEROLEMIA**	**THYROID DISEASE**	**COPD**
#1	M	44	NO	TRAUMATIC (simple fall)	large	24.7	NO	NO	NO	NO	NO	NO
#2	M	44	YES	TRAUMATIC (sport accident)	large	24.7	NO	NO	NO	NO	NO	NO
#3	M	48	NO	TRAUMATIC (car accident)	large	24.9	YES (<10 cigs/day)	NO	NO	NO	NO	NO
#4	F	48	NO	TRAUMATIC (sport accident)	small	23.6	NO	NO	NO	NO	YES	NO
#5	M	49	NO	NO TRAUMA	massive	25.5	YES (10–20 cigs/day)	NO	NO	NO	NO	YES
#6	F	47	NO	TRAUMATIC (simple fall)	small	22	NO	NO	YES	NO	NO	NO
#7	M	48	YES	NO TRAUMA	large	32.8	NO	NO	YES	NO	NO	NO
#8	M	47	NO	NO TRAUMA	large	26	NO	NO	NO	NO	NO	NO
#9	F	48	YES	NO TRAUMA	small	24.7	NO	NO	NO	NO	NO	NO
#10	M	49	NO	NO TRAUMA	small	36.5	YES (<10 cigs/day)	NO	YES	NO	NO	NO
#11	M	49	NO	TRAUMATIC (simple fall)	large	28.7	YES (10–20 cigs/day)	NO	NO	NO	NO	YES
#12	M	36	NO	TRAUMATIC (simple fall)	large	28.4	YES (10–20 cigs/day)	NO	NO	NO	NO	NO
#13	F	48	YES	NO TRAUMA	large	24.9	YES (<10 cigs/day)	NO	NO	NO	NO	NO
#14	M	45	NO	NO TRAUMA	massive	26.5	NO	NO	NO	YES	NO	NO
#15	M	49	NO	NO TRAUMA	small	26.6	NO	NO	YES	NO	NO	NO
#16	F	45	NO	NO TRAUMA	small	21.5	NO	NO	NO	NO	YES	NO
#17	F	49	NO	NO TRAUMA	small	20.5	NO	NO	NO	NO	NO	NO
#18	F	48	YES	TRAUMATIC (simple fall)	large	24.3	NO	NO	NO	NO	NO	NO
#19	F	46	NO	TRAUMATIC (simple fall)	small	21.3	NO	NO	NO	NO	NO	NO
#20	M	47	YES	TRAUMATIC (sport accident)	massive	25	YES (>20 cigs/day)	NO	NO	NO	NO	NO
#21	M	49	NO	NO TRAUMA	massive	24.6	YES (10–20 cigs/day)	NO	NO	NO	NO	NO
#22	M	49	NO	NO TRAUMA	small	31.8	NO	NO	NO	NO	NO	YES
#23	M	45	YES	TRAUMATIC (simple fall)	large	30.1	YES (10–20 cigs/day)	NO	NO	NO	NO	NO
#24	M	48	NO	NO TRAUMA	large	26.6	YES (10–20 cigs/day)	YES (Type I)	NO	NO	NO	NO
#25	M	44	YES	NO TRAUMA	small	26	YES (10–20 cigs/day)	NO	NO	YES	NO	NO
#26	M	49	YES	TRAUMATIC (sport accident)	small	30.8	NO	NO	NO	NO	NO	NO
#27	M	42	YES	NO TRAUMA	small	35.5	YES (10–20 cigs/day)	NO	NO	NO	NO	NO
#28	M	49	YES	NO TRAUMA	small	26	NO	NO	NO	NO	NO	NO
#29	F	48	YES	TRAUMATIC (simple fall)	large	44.3	YES (<10 cigs/day)	NO	YES	NO	YES	NO
#30	M	49	NO	TRAUMATIC (simple fall)	large	28	YES (>20 cigs/day)	NO	NO	NO	NO	NO
#31	F	48	YES	NO TRAUMA	small	31.5	YES (>20 cigs/day)	NO	NO	YES	NO	NO
#32	M	46	YES	TRAUMATIC (simple fall)	small	27.3	YES (10–20 cigs/day)	NO	NO	NO	NO	NO
#33	M	44	NO	TRAUMATIC (simple fall)	large	24.7	NO	NO	NO	NO	NO	NO
#34	M	44	YES	TRAUMATIC (sport accident)	large	24.7	NO	NO	NO	NO	NO	NO
#35	M	48	NO	TRAUMATIC (car accident)	large	24.9	YES (<10 cigs/day)	NO	NO	NO	NO	NO
#36	F	46	NO	TRAUMATIC (simple fall)	small	21.3	NO	NO	NO	NO	NO	NO
#37	M	47	YES	TRAUMATIC (sport accident)	massive	25	YES (>20 cigs/day)	NO	NO	NO	NO	NO
#38	M	36	NO	TRAUMATIC (simple fall)	large	28.4	YES (10–20 cigs/day)	NO	NO	NO	NO	NO
#39	F	48	YES	NO TRAUMA	large	24.9	YES (<10 cigs/day)	NO	NO	NO	NO	NO
#40	M	47	NO	NO TRAUMA	large	26	NO	NO	NO	NO	NO	NO
#41	F	48	YES	NO TRAUMA	small	24.7	NO	NO	NO	NO	NO	NO
#42	M	49	NO	NO TRAUMA	small	36.5	YES (<10 cigs/day)	NO	YES	NO	NO	NO
#43	M	49	NO	TRAUMATIC (simple fall)	large	28.7	YES (10–20 cigs/day)	NO	NO	NO	NO	YES
#44	M	49	NO	NO TRAUMA	massive	24.6	YES (10–20 cigs/day)	NO	NO	NO	NO	NO
#45	M	49	NO	NO TRAUMA	small	31.8	NO	NO	NO	NO	NO	YES
#46	M	45	NO	NO TRAUMA	massive	26.5	NO	NO	NO	YES	NO	NO
#47	M	49	NO	NO TRAUMA	small	26.6	NO	NO	YES	NO	NO	NO
#48	F	45	NO	NO TRAUMA	small	21.5	NO	NO	NO	NO	YES	NO
#49	F	49	NO	NO TRAUMA	small	20.5	NO	NO	NO	NO	NO	NO
#50	F	48	YES	TRAUMATIC (simple fall)	large	24.3	NO	NO	NO	NO	NO	NO
#51	F	48	NO	TRAUMATIC (sport accident)	small	23.6	NO	NO	NO	NO	YES	NO
#52	M	49	NO	NO TRAUMA	massive	25.5	YES (10–20 cigs/day)	NO	NO	NO	NO	YES
#53	F	47	NO	TRAUMATIC (simple fall)	small	22	NO	NO	YES	NO	NO	NO
#54	M	48	YES	NO TRAUMA	large	32.8	NO	NO	YES	NO	NO	NO
#55	M	45	YES	TRAUMATIC (simple fall)	large	30.1	YES (10–20 cigs/day)	NO	NO	NO	NO	NO
#56	M	48	NO	NO TRAUMA	large	26.6	YES (10–20 cigs/day)	YES(Type I)	NO	NO	NO	NO
#57	M	44	YES	NO TRAUMA	small	26	YES (10–20 cigs/day)	NO	NO	YES	NO	NO
#58	M	49	YES	TRAUMATIC (sport accident)	small	30.8	NO	NO	NO	NO	NO	NO
#59	M	42	YES	NO TRAUMA	small	35.5	YES (10–20 cigs/day)	NO	NO	NO	NO	NO
#60	M	49	YES	NO TRAUMA	small	26	NO	NO	NO	NO	NO	NO
#61	F	48	YES	TRAUMATIC (simple fall)	large	44.3	YES (<10 cigs/day)	NO	YES	NO	YES	NO
#62	M	49	NO	TRAUMATIC (simple fall)	large	28	YES (>20 cigs/day)	NO	NO	NO	NO	NO
#63	F	48	YES	NO TRAUMA	small	31.5	YES (>20 cigs/day)	NO	NO	YES	NO	NO
#64	M	46	YES	TRAUMATIC (simple fall)	small	27.3	YES (10–20 cigs/day)	NO	NO	NO	NO	NO

**Table 3 medicina-59-00998-t003:** Correlation between rotator cuff tear size and age; mechanism of lesion; and body mass index (BMI). SD: Standard deviation.

	Tear Size			
	*Small*	*Large*	*Massive*	*p* Value
Age (SD)	47.20 (2.18)	46.31 (3.54)	47.50 (1.91)	0.632
Mechanism of lesion				
● *No trauma*	20	8	6	0.481
● *Traumatic* (simple fall)	6	14	0	
● *Traumatic* (sport accident)	4	2	2	
● *Traumatic* (car accident)	0	2	0	
Smoking habit				
● *NO*	20	8	4	0.432
● *YES: <10 cigs/day*	2	6	0	
● *YES: 10–20 cigs/day*	6	10	2	
● *YES: >20 cigs/day*	2	2	2	
BMI				
	27.04	28.34	25.40	0.562

## Data Availability

Not applicable.

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
