# Peer review of "Aetiopathogenesis of Rotator Cuff Tear in Patients Younger than 50 Years: Medical Conditions Play a Relevant Role"

_medicina, 2023, doi:10.3390/medicina59050998_

Round 1

Reviewer 1 Report

please find attached

Reviewer 2 Report

Thank you for your effort on the paper. 

- In abstract section and in text, before giving the abbreviations give the full name. (e.g. COPD)

-Why did you exclude the patients with subscapularis tear?

- Please give detailed information about which tendons teared in type 4 tears

-At the end of the first paragraph of the discussion section, please state the most important findings of your study.

-There is no limitation section in your discussion. Please provide limitations of your study.

Reviewer 3 Report

The aim of the paper is not completely clear. Through a retrospective design, the authors attempt to demonstrate that rotator cuff tears are not primarily traumatic. At the same time, they try to investigate the etiopathogenesis of the tear. The article’s wording is unclear, it does not have clear objectives or hypotheses and it seems that the conclusions do not fully agree with the findings.

Round 2

Reviewer 1 Report

Thank you for your responses and adaptations. Nevertheless, several concerns remain.

For example,

-       Original main comment 1: "it is not clear which rotator cuff tears the authors are referring to and why they excluded subscapularis tears", the authors have added a sentence in the limitation section although this is a comment that should be addressed throughout the entire manuscript and it should be clear from the introduction which rotator cuff tears you are referring to.

-       Main comment 2: "further clarification to demonstrate that no ethical approval was needed", the justification of the authors is not sufficient. The participants data presented in this publication could allow to identify the respective patients therefore jeopardizing anonymity. A confirmation statement from the ethical committee is required to confirm that the authors indeed do not need ethical approval.

-       The description of how matched data was retrieved is incomplete, the authors should state where they found the data, how they found it, how they got access to it etc. 

Reviewer 3 Report

I have read the the text a I think the paper is suitable for publication.
